# Effects of Exergames and Protein Supplementation on Body Composition and Musculoskeletal Function of Prefrail Community-Dwelling Older Women: A Randomized, Controlled Clinical Trial

**DOI:** 10.3390/ijerph18179324

**Published:** 2021-09-03

**Authors:** Simone Biesek, Audrin Said Vojciechowski, Jarbas Melo Filho, Ana Carolina Roos de Menezes Ferreira, Victória Zeghbi Cochenski Borba, Estela Iraci Rabito, Anna Raquel Silveira Gomes

**Affiliations:** 1Postgraduate Program in Physical Education, Federal University of Paraná, Curitiba 81531-900, PR, Brazil; asaidvoj@gmail.com (A.S.V.); jarbasmf@hotmail.com (J.M.F.); 2Postgraduate Program in Food and Nutrition, Nutrition Department, Federal University of Paraná, Curitiba 81531-900, PR, Brazil; anacarolinaroos@hotmail.com (A.C.R.M.F.); estelarabito@yahoo.com.br (E.I.R.); 3Postgraduate Program in Internal Medicine, Internal Medicine Department, Endocrinology and Metabology Service (SEMPR), Hospital de Clínicas, Federal University of Paraná, Curitiba 80030-110, PR, Brazil; vzcborba@gmail.com; 4Postgraduate Program in Physical Education, Prevention and Rehabilitation in Physiotherapy Department, Federal University of Paraná, Curitiba 81531-900, PR, Brazil

**Keywords:** frail older, body composition, musculoskeletal system, virtual reality, nutritional supplementation, fatigue

## Abstract

This study aimed to investigate the effects of exergames and protein supplementation on the body composition and musculoskeletal function of pre-frail older women. Methods: A randomized controlled clinical trial was conducted with 90 pre-frail older women (71.2 ± 4.5 years old) divided into five groups: control (CG); exergames training (ETG); protein supplementation (PSG); exergames combined with protein supplementation (ETPSG); exergames combined with isoenergetic supplementation (ETISG). The primary outcomes were pre-frailty status, body composition (appendicular muscle mass (ASM); appendicular muscle mass index (ASMI)) assessed by dual energy X-ray absorptiometry and gastrocnemius muscle architecture via ultrasound. Secondary outcomes were protein intake, plasma levels of interleukin (IL)-6, plantar and dorsiflexion isokinetic peak torque, and handgrip strength (HS). Data were analyzed using an ANOVA mixed model test and Bonferroni post hoc test (*p* < 0.05). The ETG showed a reduction of ASM (16.7 ± 3.4 vs. 16.1 ± 3.3 kg; Δ = −0.5; *p* = 0.02; *d* = 0.26) and ASMI (6.8 ± 0.9 vs. 6.5 ± 0.9 kg; Δ = −0.2; *p* = 0.03; *d* = 0.35), without changing ASM in other groups. The average protein intake in the supplemented groups (PSG and ETPSG) was 1.1 ± 0.2 g/kg/day. The dorsiflexion peak torque increased 11.4% in ETPSG (16.3 ± 2.5 vs. 18.4 ± 4.2 Nm; *p* = 0.021; *d* = −0.58). The HS increased by 13.7% in ETG (20.1 ± 7.2 vs. 23.3 ± 6.2 kg, Δ = 3.2 ± 4.9, *p* = 0.004, *d* = −0.48). The fatigue/exhaustion reduced by 100% in ETG, 75% in PSG, and 100% in ETPSG. Physical training with exergames associated with protein supplementation reversed pre-frailty status, improved the ankle dorsiflexors torque, and ameliorated fatigue/exhaustion in pre-frail older women.

## 1. Introduction

Frailty represents a state of age-related physiological vulnerability, which results from the body’s reduced ability to cope with adverse health situations, such as hospitalizations, falls, and functional reduction [1,2], with a high prevalence in women [3,4]. According to the frailty phenotype, a pre-frail older person presents one or two of the following criteria: unintentional weight loss, muscle weakness, exhaustion, low physical activity, and slowness [5]. Factors such as chronic inflammation, obesity, muscle quality, physical inactivity, and inadequate nutrition may culminate in adverse outcomes in the musculoskeletal function of the physically frail older adults [6,7,8].

Multi-domain interventions (a combination of more than one intervention, e.g., exercise and nutrition interventions) [9,10,11,12], as well as multi-component physical training (e.g., a combination of resistance exercise, aerobics, balance, and flexibility) [13], are recommended to manage physical frailty in older adults.

Exergames emerge as a safe and viable possibility for physical training in the frail older population due to the benefits in balance and mobility and muscle mass gain, mainly when associated with resistance training [14,15,16]. In community-dwelling older women, exergames show high adherence and compliance to physical training. There were gains in both knee and ankle muscle strength, quadriceps cross-sectional area, and gastrocnemius architecture, which are essential aspects to improve functionality, and to prevent sarcopenia and physical frailty [17,18,19].

Higher protein intake also appears to reduce the risk of frailty in women [20] and increase muscle mass, strength, and function [21,22]. Although the rationale for supplementing protein in older adults is considered strong for multiple factors, including their reduced anabolic response to protein intake and their elevated prevalence of inflammatory and catabolic conditions associated with aging [23,24], the substantial information on the association between frailty and dietary protein comes from observational studies. Randomized clinical trials in support of high protein are scanty [24], suggesting that large clinical trials in heterogeneous populations are necessary to quantify the clinical benefit of nutrition on different clinical outcomes, with and without exercise [23].

When combined with resistance and/or multi-component exercise training, higher protein intake has been more effective than single domain interventions [10], improving frailty status, lean mass, muscle strength, and mobility in frail older adults [25]. However, most studies have significant limitations, such as including frail and non-frail older people with different degrees of nutritional impairment [21,22,26] in the same cohort, variation in the amount and quality of protein provided, and not presenting the participant’s nutrient intake [11,27]. Furthermore, the effects of multi-component interventions combining progressive resistance training using virtual games with protein supplementation on reducing frailty criteria or reversing pre-frailty status and improving musculoskeletal function in community-dwelling older people have not yet been investigated.

This study hypothesized that multi-component interventions based on virtual games combined with protein supplementation improve body composition and musculoskeletal function more than single domain interventions, such as virtual game training or protein supplementation. Multi-component interventions can also reduce and/or reverse the pre-frail state reinforcing the benefits of combining interventions. Therefore, this randomized controlled trial aimed to verify the effects of progressive resistance training with exergames combined or not with protein supplementation on pre-frailty status and criteria, body composition (appendicular muscle mass (ASM); appendicular muscle mass index (ASMI); plasma levels of (IL)-6 and musculoskeletal function (strength, mass, and architecture) in pre-frail older women.

## 2. Materials and Methods

This study was a randomized controlled trial, based on the recommendations of the Consolidated Standards of Reporting Trials (CONSORT) [28] and registered on the virtual platform of the Brazilian Registry of Clinical Trials-ReBEC (RBR-73c67m). The study was approved by the Ethics Committee for Research in Human Beings of the Hospital de Clínicas da Universidade Federal do Paraná, Curitiba, Paraná, opinion number 1,804,775 to the attributions defined in the National Health Council Resolution 466/2012. The study was conducted from January 2017 to December 2018.

### 2.1. Participants and Randomization

Older women (≥65 years), who scored on one or two criteria established by the frailty phenotype [5], met the inclusion criteria, and signed an informed consent form (ICF) agreed to participate in the study. The inclusion criteria were as follows: had “moderate” kidney functioning (i.e., a glomerular filtration rate (GFR) of 30–60 mL/min/1.73 m^2^), estimated by the Chronic Kidney Disease Epidemiology Collaboration (CKD-EPI) equation; if presented, Type II diabetes had to be compensated (<8% glycated haemoglobin); and had adequate visual acuity assessed by the Snellen card (20/70 unilateral). The exclusion criteria were as follows: had acute or terminal illness; metabolic instability or decompensated cardiovascular disease; cognitive deficits as determined by the Mini-Mental State Examination, and relative to school education; neurological disorders and/or traumatic-orthopedic conditions that prevented the participant from conducting the evaluations and/or the proposed exercises; type I diabetes; taking medication that might affect muscle metabolism (e.g., corticoids) or postural balance (e.g., anticholinergics, antihistamines, benzodiazepines, calcium channel antagonists or dopamine receptor antagonists); being intolerant/allergic to milk protein; an auditory deficit that prevents understanding of verbal instructions; and any serious deficiency recorded in the medical records, such as cardiac, respiratory, or hepatic deficiency and/or decompensated arterial hypertension (BP ≥ 140/90 mmHg).

The assessment of physical frailty, medication use, and number of diseases are detailed in a previously published protocol [29]. All procedures were performed at the beginning of the study and 12 weeks later in Curitiba, Paraná, Brazil. Following inclusion, the participants were randomized into blocks (multiples of five) to form groups. The randomization sequence was performed at randomization.com (accessed on 9 March 2017).

Due to the type of physical activity intervention, the participants knew which group they were allocated to, yet they remained blinded to the supplement they were consuming. The participants were randomized to one of five parallel groups: control group (CG); exergames training group (ETG); protein supplementation group (PSG); exergames training group associated with protein supplementation (ETPSG), and exergames training group associated with isoenergetic supplementation (ETISG).

### 2.2. Interventions

#### 2.2.1. Supplementation

After randomization, the participants selected to use the supplements received orientation on preparing the product and the quantity to use. The amount indicated for Monday to Friday was a daily dose of 6 measuring scoops per day (42 g powder) diluted in 160 mL of filtered water, for a final volume of 200 mL, with vanilla flavor, for 12 weeks. The offered 42 g serving contained 171 kcal and 21 g of whey protein isolate, 10 g of carbohydrates, 5 g of lipids, 224 mg of calcium, 3.3 mcg of vitamin D, 23 mg of vitamin C, 2300 mg of leucine, and 12 g of essential amino acids in the serving (Bemmax^®^), provided by Prodiet Medical Nutrition company (Curitiba, Brazil)

The participants were instructed to take the supplement in their mid-morning, afternoon, or before bedtime snacks. For the older women in the training group consuming the protein supplement, the instruction was to use the supplement up to the first hour after training, whenever possible.

For the isoenergetic supplement training group (ETISG), the participants received an approximate amount of the supplement (i.e., five measures of 7 g (35 g)) with an isoenergetic amount (150 kcal) similar to the protein supplement. Each measure (7 g) provided 30 kcal. The product contained 100% maltodextrin (CarboCH^®^), produced by Prodiet Medical Nutrition. The goal of the ETISG was to offer the same caloric amount as the PSG and ETPSG, yet to minimize the effect of the caloric addition on the supplemented groups.

The average consumption concerning what was offered was considered to verify adherence to the prescribed oral supplement. Consumption was considered adequate when >75% and inadequate when less than or up to 75% of the prescribed amount during the 12 weeks of follow-up

A calendar was provided where the participant was recommended to mark with an “x” all the days she would use the product to help subjects adhere to the supplement’s use. Acceptance was stratified into three categories, being A = intake less than or equal to 50% compared to the prescribed amount; B = intake between 50.1% to 74.9%, and C = intake greater than or equal to 75% compared to the product prescription [30].

#### 2.2.2. Physical Training

The exercises with Exergames consisted of Nintendo Wii Fit Plus^®^ (Nintendo Company, Minami-ku-based, Kyoto, Japan) games, using the balance platform. Physical training was performed twice a week, for approximately 50 min, with direct supervision by professional physiotherapists and physical education professionals (one per participant), for 12 weeks. The training was divided into a warm-up (approximately 10 min), neuromotor exercises (approximately 10 min), resistance exercises (approximately 20 min), and a cool-down (approximately 10 min). The games used in each training modality, as well as the progression of neuromotor and strength exercises are described in a previously published protocol [29]. The resistance exercises were performed using a weighted vest. For the first 2 weeks, the physical training participants performed the exercises using a weighted vest with no additional weight. Starting from the third week, the vest load was increased with 5% of each participant’s body mass. Progression of load occurred every two weeks with an additional 1–2% of the body mass, according to the mass measured in that week.

### 2.3. Primary Outcomes

#### 2.3.1. Pre-Frailty State

Fried’s frailty phenotype was assessed in the participants after 12 weeks of the study to evaluate if the interventions reversed, reduced, or changed pre-frailty status and criteria [5].

#### 2.3.2. Body Mass and Composition

Body mass and height were measured with Filizola^®^ mechanical scales (Filizola, São Paulo, Brazil) and Tonelli Gomes^®^ stadiometer (Tonelli^®^, Criciúma, SC, Brazil), according to the Food and Nutrition Surveillance System of the Brazilian Ministry of Health recommendations [31]. Body mass index (BMI) was calculated by dividing body mass by height squared (kg/m^2^). Waist and calf circumference were also measured. The methodological details are described in a published protocol [29]. Lean mass (MM-kg), lower (lower limbs MM), and upper (upper limbs MM) were analyzed. Limb fat mass (kg), total body fat in percentage (%), android and gynoid fat in percentage (%), and fat mass in lower and upper limbs (kg) were also considered. Appendicular skeletal muscle mass (ASM) was obtained by summing the lean mass of lower (legs) and upper (arms) limbs in kg. The sum of lower and upper limb skeletal muscle mass was divided by the participant’s height squared (ASMI = (Lean leg mass (kg) + lean arm mass (kg))/height (meters)^2^ to calculate the appendicular skeletal muscle mass index (ASMI). Body composition was analyzed by dual-energy X-ray absorptiometry (DXA) using Lunar Prodigy Advance equipment (PA+302284, Lunar Radation Corp., Madison, WI, USA) and Encore software (version 12.10.113, 2008) (GE Medical Systems Lunar, Madison, WI, USA) with a standard error of 1.6%. Precision error for body composition was 0.87 for bone mineral density (BMD), 2.32% for total fat mass, 1.67% for total lean mass, and 1.56% for percent body fat. The examination followed the International Society for Clinical Densitometry (ISCD) recommendations.

#### 2.3.3. Muscle Architecture

Muscle architecture (comprised of fascicle length (FL), pennation angle (PA), and muscle thickness (MT) of the medial gastrocnemius (GM) muscle) was assessed using B-mode ultrasound (LogiqBookXP, GE^®^) (GE Healthcare, Little Chalfont, UK) and linear array transducer (3.83 mm, 11 MHz, GE^®^). Only the right leg GM muscle was assessed, with the participant lying in the prone position, with fully extended knees and relaxed muscles, feet off the stretcher, and ankle joint at 115°, which corresponds to the angle of repose of the tibiotarsal joint [18]. Three images were collected at fixed depth of 4 cm at 30% and 40% of the total distance between the popliteal line and the medial malleolus. The probe was positioned longitudinally to the muscle fibers and coated with water-soluble transmission gel, which provided acoustic contact without depressing the dermal surface [18]. MT was defined as the mean distance between deep and superficial fascial planes, measured at five places along the ultrasound image [18,32]. The cutoff point of ≤1.5 cm was adopted to consider reduced muscle thickness [33]. PA was defined as the angle of insertion of muscle fiber fascicles into the deep fascial plane [34] and FL was defined as the length of the fascicular into the superior and deep fascial planes. When the end of the fascicle extended off the acquired ultrasound images, FL was estimated by a trigonometric function [35].

To ensure that all pre- and post-intervention measurements were taken at the exact location, a sheet of transparent acetate was drawn up as a base map indicating the locations for transducer positioning (marks on the participant’s leg, as well as the points marked for measurement) at baseline. This map was repositioned on the participant’s leg after 12 weeks of intervention [18]. All images were analyzed with Image J software (Version 1.46r) (National Institutes of Health, Bethesda, MD, USA). A single, previously trained researcher performed the GM evaluations and measurements, and the intraclass correlation (ICC) was calculated for the MT (ICC = 0.8), PA (ICC = 0.6), and FL (ICC = 0.8) measurements.

### 2.4. Secondary Outcomes

#### 2.4.1. Biochemical Tests and Plasma Quantification of IL-(6)

Blood samples were collected after an eight-hour fast. Glycemic control was performed by measuring glycated hemoglobin (A1c) using the liquid chromatography-HPLC method. From the serum creatinine values, the glomerular filtration rate was estimated by the Chronic Kidney Disease Epidemiology Collaboration (CKD-EPI) equation [36]. The concentrations of 25 dihydroxy-vitamin D (ng/mL) were analyzed by chemiluminescence. Plasma interleukin (IL)-6 pg/mL levels were analyzed by the electrochemiluminescence method to monitor inflammation. A cutoff point greater than 1.4 pg/mL was used to characterize high levels of interleukin-6 [37].

#### 2.4.2. Peak Torque and Handgrip Strength

Muscle strength was assessed using an isokinetic dynamometer (BIODEX, System 4 Pro™ model) (Biodex, Corp., Shirley, NY, USA) in concentric form for two leg muscle groupings: plantar flexors and ankle dorsiflexors. Participants performed two repetitions for each test: a submaximal series for familiarization and a maximal series for registration, encouraged by voice command from a single experienced assessor. Five repetitions were performed for the concentric test, at 60°/s and 180°/s speed, for the analysis of the plantar flexors and ankle dorsiflexors. The mode of execution of the movements was concentric/concentric. The participant was asked to perform five submaximal repetitions to be familiarized with the movement. After a one-minute rest, five repetitions of the maximal test for recording were performed. The total range of motion for the test was 40 degrees for the ankle joint (starting from 10° dorsiflexion to 30°) [38]. The data were analyzed by peak torque (PT). Handgrip strength (HS) was also evaluated according to a published protocol [29].

#### 2.4.3. Food Consumption

Food intake was assessed by the three-day food record method (R3d). The methodological details are described in a previously published protocol [29].

### 2.5. Sample Size

The sample size was 80 participants, with an addition of 10% for possible sample loss, resulting in 90 participants, 18 in each group. The sample size was calculated using the GPower 3.19 program (Heinrich-Heine-Universität Dusseldorf, Dusseldorf, Germany), in the F-test family. Further, a repeated measures ANOVA statistical test for within- and between-group interactions was utilized. This is an a priori analysis type with a convention effect size of 0.4 (large), type I error of 0.05, and power of analysis of 0.8.

### 2.6. Statistical Analysis

The normality of the data distribution was assessed using the Kolmogorov-Smirnov test. Parametric data were expressed as mean (±standard deviation) and non-parametric data as median (minimum; maximum) and absolute and relative frequency when categorical. For comparisons of categorical data, the Chi-square and Fisher’s Exact tests were applied. For within- and between-group comparisons and their interactions, an ANOVA (mixed model) followed by a Bonferroni’s post hoc test were used to check for differences in the main effects of the factors time (pre and post) and groups (CG, ETG, PSG, ETPSG, and ETISG) and the interactions among these factors. For non-parametric variables, the Kruskal–Wallis test was used, followed by Wilcoxon’s signaled ranks test and McNemar’s test. The within group and between-group effect size (ES) was also calculated using Cohen’s *d* equation for dependent (considering the Pearson correlation coefficient: *r*; *d* = t _dependent_√2(1 − r)/n), and independent samples (*d* = t _independent_ √n_1_ + n_2_/n_1_ × n_2_). The effect sizes were considered “small,” “medium,” and “large” for r = 0.2, 0.5, 0.8, respectively [18,39]. All analyses were performed using the SPSS program (version 22^®^) (SPPS Inc, Chicago, IL, USA) considering a significance level of 95% (*p* < 0.05).

## 3. Results

Of the 272 older women assessed, 182 older women were excluded from the study because they did not meet the inclusion criteria. Ninety older women with a mean age of 71.2 ± 4.5 years, classified as pre-frail according to the frailty phenotype, were randomized. Regarding the frailty criteria at the pre-intervention time point, 50% (*n* = 45) scored reduced strength (HS), followed by fatigue/exhaustion (38.8%, *n* = 35), weight loss (15.5%, *n* = 14), low energy expenditure (3.3%, *n* = 3), and slowness (3.3%, *n* = 3). As for the number of frailty criteria, 72.2% (*n* = 65) scored on one criterion for frailty and 27.7% (*n* = 25) scored on two criteria. All other investigated characteristics showed no significant difference between the five intervention groups at baseline (Table 1).

The analysis losses after intervention are depicted in Figure 1.

A significant interaction (*p* = 0.028) was observed on protein intake per kilogram weight in the CG with reduced by about 22% after 12 weeks compared to the pre-time point (1.1 g/kg vs. 0.9 g/kg; *p* = 0.015; *d* = 0.81). In addition, there was a significant increase in the percentage of protein intake between groups (*p* = 0.000) of 17.1% vs. 20.1% in PSG. A significant interaction (*p* = 0.049) was also observed on lipid intake in the CG, with a 33.5% reduction in total fat intake. In relation to energy intake, we observed a difference between groups following the experiment. There was a 23.8% reduction in energy intake from 1766.7 kcal to 1426.5 kcal (*p* = 0.004, *d* = 1.07) in the CG and 12.7% increase in total energy intake in the PSG from 1504.6 kcal to 1723.1 kcal (*p* = 0.001, *d* = −0.60). As for total protein intake (g/day), a significant 22% increase was observed in the PSG of 63.9 g vs. 82.9 g (*p* = 0.006, *d* = −1.45) and 28% increase in the ETPSG of 58.1 g vs. 81.5 g (*p* = 0.06, *d* = −0.84) between groups post experiment was also perceived. Furthermore, a 9.6% increase in carbohydrate intake was observed within groups in the ETISG (*p* = 0.013, *d* = −0.78) (Table 2).

The adherence rate to the use of protein and/or isoenergetic supplementation was classified as Intake B, with average supplement intake ranging from 50.1% to 74.9% of that prescribed.

In comparisons of body composition analyzed by DXA, we observed a significant 3.5% reduction in android fat in within-group comparisons (*p* = 0.024) in the PSG after the intervention (44.3 ± 7.1 vs. 42.8 ± 6.4%, Δ = −0.4 ± 0.7%; *p* = 0.045; *d* = 0.21). We also observed a significant difference between groups (*p* = 0.001) and interaction (*p* = 0.001) in the Up lim MM (kg) with a 2.8% reduction in ETG (3.7 ± 0.9 vs. 3.6 ± 0.9 kg; Δ = −0.1 ± 0.2 kg; *p* = 0.002, *d* = 0.2), after training. A 3.7% reduction was observed in the ASM of the ETG (16.7 ± 3.4 vs. 16.1 ± 3.3 kg, Δ = −0.6 ± 0.1 kg, *p* = 0.02, *d* = 0.26) and ASMI (6.8 ± 0.9 vs. 6.5 ± 0.9 kg/m^2^, Δ = −0.3 ± 0.0, *p* = 0.03, *d* = 0.35) after physical training. No significant difference was observed in the other variables for within group, between group, and interaction variables (Table 3).

There was no significant difference in any variables when comparing within- and between-group, interactions, and medial gastrocnemius muscle architecture (Table 4).

Concerning the plasma levels of (IL)-6, there was no significant difference in any of the comparisons. A within-group difference was observed for PT of the ankle dorsiflexors at 60°/s ankle (*p* = 0.031), with an 11.4% increase in ETPSG (16.3 ± 2.5 vs. 18.4 ± 4.2 Nm, Δ = 2.13 ± 3.4, *p* intragroup = 0.021, *d* = −0.58). There was a significant change in the HS from pre- to post-intervention (*p* = 0.008), with a 13.7% increase in ETG (20.1 ± 7.2 vs. 23.3 ± 6.2 kg, Δ = 3.2 ± 4.9, *p* intragroup = 0.004, *d* = −0.48). However, no significant difference was observed at the two time points (pre- and post-intervention) when evaluated between intervention groups (Table 5).

There was a significant reduction in the number of participants with exhaustion criterion, being 100% in the ETG (pre: *n* = 7, 46.7% versus. post: *n* = 0, 0.0%, *p* = 0.016), 75% in the PSG (pre: *n* = 8, 44.4% vs. post: *n* = 2, 11.1%, *p* = 0.031), and 100% in the ETPSG (pre: *n* = 7, 43.8% versus. post: *n* = 0, 0.0%, *p* = 0.016) groups. No significant difference was observed in other frailty criteria both pre- and post-experiment. Regarding the frailty classification, in CG, seven subjects (46.7%) changed from pre-frail to non-frail and one (6.7%) from pre-frail to frail (*p* = 0.016); in ETG, 11 (73.3%) participants changed status from pre-frail to non-frail (*p* = 0.001); in ETPSG, 10 (55.6%) changed from pre-frail to non-frail and 1 (5.6%) from pre-frail to frail (*p* = 0.000); in ETISG, seven (43.8%) changed from pre-frail to non-frail (*p* = 0.016) (Table 6).

## 4. Discussion

Physical training exergames associated with protein supplementation (WiiProtein) reversed the pre-frailty state, reduced fatigue/exhaustion, and increased ankle dorsiflexors torque without difference between groups. The single domain intervention with exergames increased the handgrip strength, decreased ASM and ASMI (within group), and reduced the Up Lim MM (kg) in relation to the other groups, although this difference was subtle when compared to the other intervention groups. Meanwhile, the single domain with protein supplementation decreased android fat, without significant difference in the muscle mass in prefrail community-dwelling older women.

Other studies that evaluated multi-domain interventions, combining physical training and calorie-protein [11] supplementation or nutrition education [12], also found a reduction in the criteria and prevalence of physical frailty. However, none of these studies mentioned which frailty criteria were improved after interventions. Muscle weakness and self-reported fatigue/exhaustion are two Frailty Phenotype criteria frequent in frail and pre-frail older, and the most prevalent in the present study [11,40,41]. Even without difference when compared to the control group, we must highlight the reduction of fatigue/exhaustion because the majority of participants were included reporting this criterion. Thus, when we found, at post intervention (exergames or protein supplementation or the association of both), that fatigue/exhaustion had ameliorated and this outcome was not observed in the control group, we suggest an important clinical relevance of WiiProtein, isolated or in combination. We might suppose that multifactorial intervention (WiiProtein) should be used for pre-frail older women while a reduction in fatigue/exhaustion and increase in muscle strength are associated with delayed progression of frailty, as detected in the present study with pre-frailty reversion [42].

The WiiProtein interventions were not different from control group, instead of improvements more pronounced in the dorsiflexion torque. The study demonstrated the importance of protein supplementation, alone or associated with exercise and social integration during the study in the management of pre-frailty in women. Thus, even if older women were unable to exercise, protein supplementation alone was sufficient to improve fatigability, preventing negative prognosis.

Multi-component physical training, especially with the addition of progressive resistance, is a strategy to improve muscle strength, physical performance, balance, and flexibility, as well as to reverse the state of frailty in pre-frail and frail older people [13,43,44]. Interventions associated with protein supplementation can favor better outcomes [9]. Consistent with these findings, in our study, only the pre-frail older women in the ETG and ETPSG had muscle strength gains when compared to other intervention groups. However, the combination of exergames and protein supplementation resulted in the maintenance of lean body mass without additional gains. Contrarily, exercise training without supplementation demonstrated a decline in Up lim MM (kg) and ASM mass after intervention. One study observed an increase in strength and muscle mass in frail older people after resistance training intervention associated with 30 g of protein supplementation for 24 weeks [22]. This study suggests that frail older people under resistance training should take protein supplementation to increase skeletal muscle mass. Therefore, we did not observe an increase in lean body mass nor changes in muscle architecture in our non-intervention group. This may be associated with a low supply of protein supplementation (21 g/day), or a training protocol with exergames, which may have been insufficient in frequency (2×/week) and duration (12 weeks), since some studies have identified benefits when the physically frail perform physical training three times per week, for 30–45 min/session, for at least 16 weeks [45].

The physical training with virtual games proposed in this study emphasize on exercises for the lower limbs, with focus on the gastrocnemius muscle (e.g., palm tree, chair), with a progressive increase in load (resistance) of the exercises every two weeks, in addition to the stimulus through protein supplementation. Nevertheless, no changes in muscle architecture were verified after 12 weeks of intervention. Even though the muscle architecture presented no change, by assessing muscle architecture using ultrasonography (US), it was found that pre-frail older women had reduced MT and FL of the medial gastrocnemius at the pre-intervention time point. This factor indicates sarcopenia, considering the cutoff points of 1.5 cm for MT found in sarcopenic older individuals [33,46]. A study conducted with community-dwelling older women who underwent physical training with virtual games found a 1.3% increase in the quadriceps cross-sectional area [17]. The mentioned study used magnetic resonance imaging, a non-invasive method that is considered the gold standard for assessing muscle quality and quantity [47]. Other studies verified an 8.7% increase in the thickness of the medial gastrocnemius muscle after 12 weeks of virtual dance training performed three times a week in community-dwelling older women [18]. Further, there were gains in MT, PA, and FL of the medial gastrocnemius after training for eight weeks with ballroom dancing (three times per week for 60 min) [48].

Regardless of the investigated muscle architecture changes, the ETPSG participants increased the dorsiflexors’ torque, reinforcing the non-linearity of the relationship between muscle mass and strength, and suggesting neural gains [49]. The improvement in ankle dorsiflexion strength may have resulted from the use of an overload vest beginning in the third week of physical training, with load progression every two weeks. In addition, there was a probable effect of lower limb resistance exercises (e.g., palm tree, chair) and neuromotor exercises (e.g., balance bubble, snowboard slalom). We also assume that the physical training associated with protein supplementation contributed to the positive results, although no muscle mass gains were observed in the Lo lim MM (kg). Once, despite a significant effect, an increase in the strength of the plantiflexors was observed in the training groups associated with protein as isocaloric supplementation. Other studies investigating the effects of physical training through virtual games also found gains in Lo lim MM (kg) strength in both pre-frail community-dwelling older women [16] and moderately active community-dwelling older women [18]. Santos et al. (2019) attributed the results to the progression of the vest’s load used by participants. Gallo et al. (2019) verified a 16.3% increase in ankle plantar flexor peak torque, as well as an increase in muscle mass detected by ultrasonography (US) of the gastrocnemius muscle and calf circumference measurement after 12 weeks of physical training (three times per week) with virtual dance (Xbox 360^®^) in community-dwelling older women. However, these studies lacked an assessment of protein intake and performance of the nutritional intervention.

Therefore, we assume that the protein supply in this study probably was insufficient to stimulate protein synthesis and generate additional gains in gastrocnemius muscle thickness, especially when considering older women with likely an excess in intramuscular fat. Although protein supplementation raised protein intake by 23% in the PSG group and 29% in the ETPSG group, mean protein intake after intervention (1.1 ± 0.2 g/kg/d) was below expected (≥1.2 g/kg/day) [50] in both groups. In addition, as the pre-frail older women were overweight, the average amount of protein offered was 0.28 g/kg/day in only one additional meal. However, it was recommended that the subjects have at least three meals containing 0.4 g of protein/kg of body mass to maximize myofibrillar protein synthesis [51].

Nonetheless, increasing a dose of at least 20 g of protein/day appeared to represent a promising strategy to prevent and treat sarcopenia in the older population [52], considering that older individuals have difficulty adhering to large doses of protein supplementation. This strategy also contributes to lower body adiposity [53,54]. Our data showed that protein supplementation reduced android fat mass in PSG participants by 3.5% and preserved muscle mass.

Our data showed that protein supplementation reduced android fat mass in PSG participants by 3.5%. In addition to the excess body adiposity found in the study participants, they had prediabetes and higher cytokine level, which was observed by increased glycated hemoglobin levels (A1c) and IL-6, respectively. Thus, insulin resistance can be stimulated by the inflammatory response, which could trigger increased intramuscular fat, negatively affecting musculoskeletal function over time [55]. Therefore, protein supplementation could be advantageous for older women in order to preserve muscle mass, improve insulin action, and reduce the development and worsening of chronic diseases such as Type II diabetes [22]. In our study, the aim of IL-6 assessment was to investigate the effects of multidomain intervention on factors that could intervene in the body composition and muscle function. As previously reported, lower muscle mass and strength have been associated to higher plasma concentrations of IL-6, to frailty, and to poor health outcomes in older adults [56,57]. Instead, although participants of the present study have presented higher IL-6 levels, no effect was found for multidomain intervention.

The adherence to supplements offered in our study was lower than observed by other studies [21,22]. Although Tieland et al. (2012) controlled the use of supplements, in a similar way to ours, the product offered in this study was presented in liquid form, which may have facilitated adherence to consumption. While Park et al. (2018) offered the supplement in powder form and their adherence to the use of the supplements was controlled by the rate of non-consumed products. The modification of eating habits in older people is a great challenge. In addition, the adaptation to the organoleptic and physical characteristics of the product and the participant’s need for portioning (preparing the product and the quantity to use) may have contributed to the lower adherence to consumption in our study. It is noteworthy that, in this study, the participants were periodically encouraged (at least once a week) to use the product properly, through verbal guidance on the days of physical training and by telephone to the participants who were not undergoing training. Moreover, all participants had recipe suggestions (book with recipes tested by the researchers, with isocaloric or protein supplements). The difficulty of adherence found in this study may approach the limitations found in clinical practice. Thus, for future studies, it is suggested to evaluate the barriers and difficulties of adherence to supplementation.

Finally, physical training with exergames, encompassing balance and progressive resistance exercises, proved to be safe and motivating for community-based pre-frail older women, as observed by the high adherence rate to the study (87.7%). The practice of exercise using exergames has been classified as fun and motivating by aged individuals. Thus, the type of physical training chosen, exergames, per se, stimulates adherence. In addition, interventions with more than one domain (e.g., physical training and nutritional supplementation) appear promising in treating physical frailty

### 4.1. Strengths and Limitations

The present study has relevant strengths. It is a randomized controlled clinical trial, conducted only with community women who were classified as pre-frail. The assessment of participants’ food intake followed a standardized and recommended methodology. Participants showed good adherence to the study, and adherence to physical training with exergames and the supplement, and there were no reports of adverse effects with the use of the protein supplement. Finally, the study had a low dropout rate from the interventions. Limitations of the study include the 12-week intervention period, as well as frequency and progression of resistance, which may have been insufficient to increase the lean body mass of pre-frail older women, and the fatigue/exhaustion criterion was assessed with two questions derived from Fried’s frailty phenotype. Thus, for future research, we suggest that this criterion be evaluated by other methods including physical and cognitive function to detect the interactions between performance fatigability and perceived fatigability and by a multi-professional team, including psychology professionals. In addition, a myoelectrical analysis should be included to elucidate the mechanisms of the strength increase induced by training with virtual games on neuromuscular aspects. Finally, the study was conducted only with pre-frail women, so the results cannot be generalized to pre-frail men.

### 4.2. Implications for Clinical Practice

Our results indicated that physical training with exergames associated or not associated with protein supplementation, for 12 weeks, was able to reverse the state of frailty and reduce the report of fatigue/exhaustion after the intervention. Self-reported exhaustion appears to emerge in the development of early frailty. Furthermore, the feeling of fatigue/exhaustion can trigger behavioral changes, making individuals less active, predisposing to a vicious cycle that would direct toward frailty [58]. Therefore, early identification of this criterion through a quick and easy-to-perform instrument, such as by applying two questions taken from the Center for Epidemiologic Studies Depression Scale (CES-D), is considered a valid instrument for clinical studies [5,42,59] and should be implemented in clinical practice. Regarding the practical implications, the type of physical training can contribute to enhancing psychosocial effects if played with other players. Additionally, the benefits can be applied to cognitive functioning due to the dual task required by the virtual game. Finally, exergames were well accepted by pre-frail older women. Physical exercise using technological resources such as exergames can be a strategy to increase adherence to physical training and probably contribute to the exchange of experiences with younger family members [60].

## 5. Conclusions

Physical training with virtual games associated with protein supplementation improved the strength of dorsiflexors, and reduced exhaustion with a reversion of the pre-frailty state in pre-frail older women, while the exergame training group reduced the Up Lim MM when compared to the other groups. Therefore, multi-domain interventions seem to be the most promising for clinical practice to prevent and treat pre-frailty in community-dwelling older women.

## Figures and Tables

**Figure 1 ijerph-18-09324-f001:**
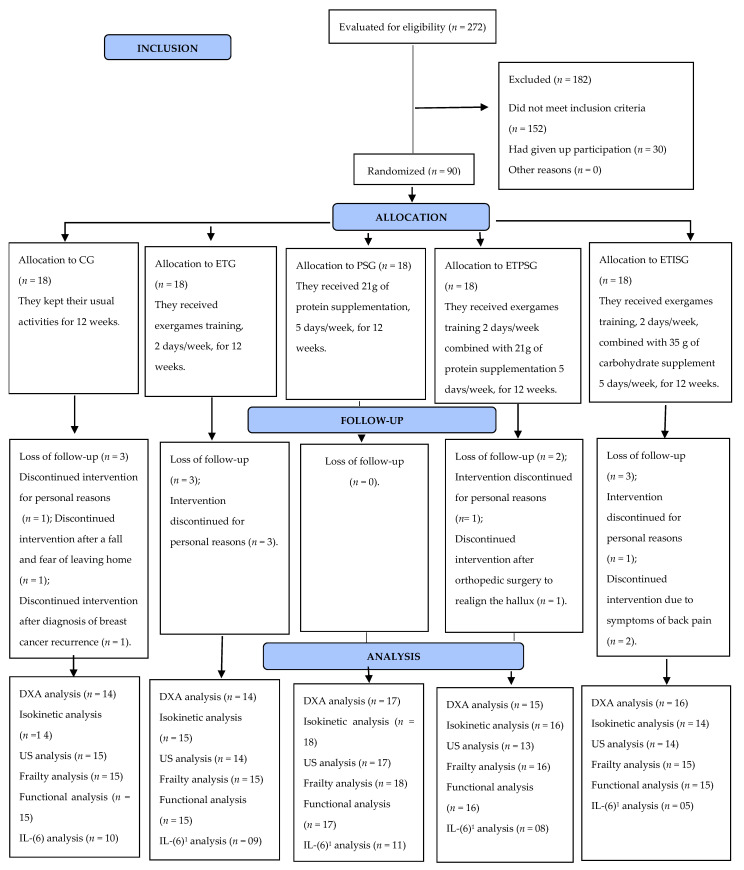
Randomization flowchart and experimental study design. CG: Control Group; ETG: Exergames Training Group; PSG: Protein Supplementation Group; ETPSG: Exergames Training combined with Protein Supplementation Group; ETISG: Exergames Training combined with Isoenergetic Supplementation Group; US: ultrasound; IL-(6): interleukin 6; DXA: dual-energy X-ray absorptiometry; ^1^: undetectable values in the analysis; *n* = number.

**Table 1 ijerph-18-09324-t001:** Characterization of the Participants Included in the Study (*n* = 90).

Variables	CG (*n* = 18)	ETG (*n* = 18)	PSG (*n* = 18)	ETPSG (*n* = 18)	ETISG (*n* = 18)	*p*
Age, mean (SD)	70.4 ± 3.9	71.2 ± 4.2	73.1 ± 5.3	71.7 ± 4.8	69.7 ± 4.0	0.203
Body mass, mean (kg)	68.4 ± 11.1	71.5 ± 13.1	67.9 ± 11.4	73.3 ± 12.3	71.6 ± 13.6	0.891
Height (m), mean (SD)	1.6 ± 0.1	1.6 ± 0.1	1.6 ± 0.1	1.6 ± 0.1	1.6 ± 0.1	0.941
BMI (kg/m^2^), mean (SD)	27.1 ± 4.3	30.1 ± 4.1	28.1 ± 3.8	30.3 ± 3.9	29.3 ± 5.6	0.582
School level, *n* (%)						
Illiterate	1 (5.6)	1 (5.6)	0 (0.0)	0 (0.0)	0 (0.0)	0.560
1–4 years	8 (44.4)	7 (38.9)	8 (44.4)	7 (38.9)	5 (27.8)	
5–8 years	3 (16.7)	3 (16.7)	6 (33.3)	5 (27.8)	2 (11.1)	
>8 years	6 (33.3)	7 (38.9)	4 (22.2)	6 (33.3)	11 (61.1)	
MMSE mean (SD)	27.3 ± 2.8	27.8 ± 2.4	27.3 ± 2.4	27.0 ± 2.6	28.4 ± 2.2	0.367
Prescription medicine number, *n* (%)	4.3 ± 2.3	3.2 ± 2.3	3.4 ± 2.6	5.5 ± 2.9	4.3 ± 2.6	0.459
Disease number, *n* (%)	2.5 ± 1.1	1.9 ± 1.3	2.3 ± 1.5	2.9 ± 1.1	2.5 ± 1.1	0.886
HbA1c (%), mean (SD)	5.9 ± 0.5	6.1 ± 0.9	6.1 ± 0.9	6.3 ± 1.4	5.9 ± 0.7	0.934
serCr (mg/dL), mean (SD)	0.9 ± 0.1	1.0 ± 0.2	0.9 ± 0.1	1.4 ± 2.0	0.9 ± 0.1	0.397
GFR (mL/min/1.73 m^2^), mean (SD)	64.5 ± 11.8	54.9 ± 10.5	67.3 ± 7.1	63.9 ± 12.9	65.5 ± 11.0	0.461
Vitamin D (ng/mL), mean (SD)	28.1 ± 6.1	27.8 ± 3.9	24.3 ± 5.7	27.1 ± 4.4	24.4 ± 3.4	0.507
Frailty Criteria						
HS reduction, *n* (%)	12 (66.7)	9 (50.0)	11 (61.1)	11 (61.1)	10 (55.6)	0.879
Exhaustion/fatigue, *n* (%)	7 (38.9)	10 (55.6)	8 (44.4)	7 (38.9)	9 (50.0)	0.823
Weight loss, *n* (%)	3 (16.7)	2 (11.1)	4 (22.2)	0 (0.0)	0 (0.0)	0.968
Caloric expenditure reduction, *n* (%)	1 (5.6)	0 (0.0)	2 (11.1)	0 (0.0)	0 (0.0)	0.505
Slowness, *n* (%)	0 (0.0)	1 (5.6)	1 (5.6)	1 (5.6)	0 (0.0)	1.000
Frailty components number						
01 component, *n* (%)	13 (72.2)	13 (72.2)	11 (61.1)	14 (77.8)	13 (72.2)	0.862
02 components, *n* (%)	5 (27.8)	5 (27.8)	7 (38.9)	4 (22.2)	5 (27.8)	

BMI: body mass index; MMSE: Mini-Mental State Examination; HbA1c: glycated haemoglobin; serCr: serum creatinine; GFR: glomerular filtration rate; HS: handgrip strength; SD: standard deviation; *n* = number.

**Table 2 ijerph-18-09324-t002:** Within- and between-group comparisons and interactions of macronutrient and energy intake.

Parametric Variables	CG (*n* = 12)	ETG (*n* = 14)	PSG (*n* = 14)	ETPSG (*n* = 12)	ETISG (*n* = 10)	p^a^	p^b^	p^c^
	Mean ± SD	Mean ± SD	Mean ± SD	Mean ± SD	Mean ± SD			
Prot Int-Pre (g/kg/day)	1.1 ± 0.4	0.9 ± 0.3	1.0 ± 0.2	0.9 ± 0.3	0.9 ± 0.3			
Prot Int-Post (g/kg/day)	0.9 ± 0.3 *	0.8 ± 0.3	1.1 ± 0.2	1.1 ± 0.2	1.1 ± 0.2	0.877	0.076	0.028
Δ	−0.2 ± 0.1	−0.1 ± 0.0	0.1 ± 0.0	0.2 ± 0.1	0.2 ± 0.1			
Prot Int-Pre (%)	17.7 ± 2.2	16.1 ± 1.5	17.1 ± 2.8	16.1 ± 2.8	17.5 ± 2.4			
Prot Int-Post (%)	17.4 ± 3.9	16.3 ± 4.6	20.1 ± 3.4 ^≠^	18.1 ± 2.1	16.4 ± 1.8	0.096	0.000	0.135
Δ	−0.3 ± 1.7	0.2 ± 3.1	3.0 ± 0.6	2.0 ± 0.7	−1.1 ± 0.6			
Lip Int-Pre (g/day)	62.9 ± 17.6	54.9 ± 19.9	58.2 ± 19.8	57.4 ± 19.8	49.2 ± 15.8			
Lip Int-Post (g/day)	47.1 ± 12.7 *	46.0 ± 14.9	60.7 ± 15.2	57.4 ± 12.0	51.5 ± 16.5	0.081	0.405	0.049
Δ	−15.8 ± 4.9	−8.9 ± 5.0	2.5 ± 4.6	0.0 ± 7.8	2.3 ± 0.7			
**Nonparametric Variables**	**Med (Min-Max)**	**Med (Min-Max)**	**Med (Min-Max)**	**Med (Min-Max)**	**Med (Min-Max)**	**p^a^**	**p^d^**	**p^e^**
Energy Int-Pre (kcal)	1766.7 (1121.6–2136.4)	1688.9 (1001.9–2275.9)	1504.6 (1106.6–2150.5)	1561.7 (1103.6–2480.1)	1477.5 (762.3–1826.4)			
Energy Int-Post (kcal)	1426.5 (677.1–1755.6) ¥	1318.9 (1167.1–2097.1)	1723.1 (1230.8–2341.3) *	1714.7 (1455.8–1995.9)	1521.4 (990.2–2282.9)	0.059	0.541	0.017
Δ	−340.2 ± 412.7	−320 ± 435.3	218.5 ± 157.5	153.0 ± 510.1	43.9 ± 239.4			
Energy Int-Pre (Kcal/kg/day)	24.8 (18.1–36.5)	20.6 (12.8–33.9)	24.2 (13.9–34.5)	19.4 (15.8–35.7)	21.7 (10.1–29.6)			
Energy Int-Post (Kcal/kg/day)	22.8 (11.7–33.2)	19.2 (12.1–30.1)	25.2 (17.1–37.0)	22.7 (16.6–36.7)	25.3 (13.0–35.6)	0.059	0.858	0.067
Δ	−2.0 ± 4,8	−1.4 ± 5.9	1.0 ± 6.8	3.3 ± 7.3	3.6 ± 3.5			
Prot Int-Pre (g/day)	72.7 (44.5–111.7)	60.8 (39.3–88.2)	63.9 (51.2–94.4)	58.1 (41.3–100.1)	59.2 (39.9–76.7)			
Prot Int-Post (g/day)	57.5 (20.6–103.6)	54.8 (30.6–124.0)	82.9 (60.6–118.0) ¥	81.5 (55.3–90.6) ¥	66.3 (41.7–86.8)	0.285	0.468	0.001
Δ	−6.0 ± 16.0	−6.0 ± 13.0	19 ± 16.5	23.4 ± 2.3	7.1 ± 5.9			
Carb Int-Pre (g/day)	186.9 (137.3–320.1)	211.6 (137.8–334.1)	191.8 (135.7–243.5)	201.4 (151.4–356.2)	197.3 (124.5–232.4)			
Carb Int-Post (g/day)	184.5 (101.4–265.1)	191.8 (146.3–295.6)	219.2 (127.4–275.7)	229.8 (195.0–293.4)	218.2 (144.4–341.4)^ #^	0.013	0.514	0.052
Δ	−2.4 ± 45.5	−19.8 ± 15.0	27.4 ± 11.9	28.4 ± 9.6	20.9 ± 64.5			

CG: Control Group; ETG: Exergames Training Group; PSG: Protein Supplementation Group; ETPSG: Exergames Training combined with Protein Supplementation Group; ETISG: Exergames Training combined with Isoenergetic Supplementation Group; kcal: kilocalories; kcal/kg/day: kilocalories per body mass per day; g/day: grams per day; g/kg/day: grams per body mass per day; p^a^: *p*-values of within-group comparisons; p^b^: *p*-values between groups; p^c^: *p*-values of interaction; p^d^: *p*-values between-group comparisons (pre-experimental); p^e^: *p*-values of between-group comparisons (post-experimental); Δ: delta (difference between pre-and post-intervention assessments); Med: Median; Min: Minimum; Max: Maximum; Int: Intake; Prot: Protein; Carb: carbohydrates; Lip: Lipids; * Significant difference interaction; ^#^:Significant difference within groups; ^≠^: Significant difference between groups; ¥: Significant difference between groups (post-experimental); *n* = number.

**Table 3 ijerph-18-09324-t003:** Within- and between-group comparisons and interactions of Body Composition.

Variables	CG (*n* = 15)	ETG (*n* = 15)	PSG (*n* = 16)	ETPSG (*n* = 16)	ETISG (*n* = 15)	p^a^	p^b^	p^c^
	Mean ± SD	Mean ± SD	Mean ± SD	Mean ± SD	Mean ± SD			
Body mass-Pre (kg)	68.4 ± 11.1	71.5 ± 13.1	67.8 ± 11.4	73.3 ± 12.3	71.6 ± 13.6	0.118	0.409	0.226
Body mass-Post (kg)	66.5 ± 12.2	72.4 ± 12.2	67.0 ± 11.3	72.5 ± 14.3	69.4 ± 10.4			
Δ	−1.9 ± 1.1	0.9 ± 0.9	−0.8 ± 0.1	−0.9 ± 2.1	−2.2 ± 3.2			
WC-Pre (cm)	96.6 ± 11.5	100.5 ± 10.2	95.5 ± 9.3	99.6 ± 9.0	97.0 ± 7.9	0.484	0.339	0.339
WC-Post (cm)	95.0 ± 10.2	101.7 ± 9.9	95.4 ± 9.9	99.2 ± 9.2	95.8 ± 8.1			
Δ	−1.6 ± 1.3	1.2 ± 0.3	−0.1 ± 0.6	−0.4 ± 0.2	−1.2 ± 0.2			
Fat mass-Pre (kg)	29.6 ± 8.3	32.6 ± 7.4	30.4 ± 7.7	33.1 ± 9.0	29.4 ± 6.9	0.308	0.594	0.541
Fat mass-Post (kg)	30.2 ± 7.3	32.2 ± 6.0	30.6 ± 7.3	33.5 ± 9.5	29.7 ± 6.3			
Δ	0.6 ± 1.0	−0.5 ± 1.4	0.2 ± 0.4	0.4 ± 0.5	0.3 ± 0.6			
Tot bod fat-Pre (%)	43.5 ± 6.2	44.9 ± 4.1	44.2 ± 5.3	44.9 ± 5.0	41.7 ± 5.4	0.365	0.393	0.656
Tot bod fat-Post (%)	43.8 ± 4.6	45.1 ± 4.3	44.1 ± 5.2	44.8 ± 5.2	41.9 ± 4.8			
Δ	0.3 ± 1.7	0.2 ± 0.2	−0.1 ± 0.1	−0.1 ± 0.2	0.2 ± 0.6			
Gynoid Fat-Pre (%)	45.2 ± 5.5	47.9 ± 6.3	48.0 ± 4.5	46.6 ± 6.5	43.7 ± 4.6	0.339	0.152	0.841
Gynoid Fat-Post (%)	44.8 ± 4.8	47.9 ± 5.9	47.6 ± 3.8	45.9 ± 6.8	43.9 ± 5.2			
Δ	−0.4 ± 0.7	0.0 ± 0.4	−0.4 ± 0.7	−0.7 ± 0.3	0.2 ± 0.6			
Android Fat-Pre (%)	45.4 ± 7.8	46.7 ± 6.1	44.3 ± 7.1	47.5 ± 5.6	43.7 ± 6.2	0.024	0.438	0.805
Android Fat-Post (%)	44.8 ± 6.9	45.8 ± 6.5	42.8 ± 6.4 ^#^	46.6 ± 5.5	43.5 ± 6.3			
Δ	−0.6 ± 0.9	−0.9 ± 0.4	−1.5 ± 0.7	−0.9 ± 0.1	−0.2 ± 0.1			
Up Lim Fat-Pre (kg)	3.4 ± 1.1	3.5 ± 0.9	3.2 ± 0.8	3.6 ± 0.9	3.7 ± 1.2	0.905	0.246	0.601
Up Lim Fat-Post (g)	3.3 ± 0.8	3.4 ± 0.9	3.2 ± 0.9	3.5 ± 1.1	3.4 ± 0.8			
Δ	−0.1 ± 0.3	−0.1 ± 0.0	0.0 ± 0.1	−0.1 ± 0.2	−0.3 ± 0.4			
Lo Lim Fat-Pre (kg)	10.0 ± 3.1	10.8 ± 1.9	10.4 ± 2.5	11.6 ± 3.9	10.8 ± 4.1	0.527	0.068	0.053
Lo Lim Fat-Post (kg)	10.2 ± 2.8	11.5 ± 3.7	10.8 ± 3.0	12.1 ± 4.4	10.5 ± 3.1			
Δ	0.2 ± 0.3	0.7 ± 1.8	0.4 ± 0.5	0.5 ± 0.5	0.3 ± 1.0			
Up Lim MM-Pre (kg)	3.5 ± 0.7	3.7 ± 0.9	3.4 ± 0.6	3.7 ± 0.6	3.9 ± 0.8	0.306	0.001	0.001
Up Lim MM-Post (kg)	3.4 ± 0.7	3.6 ± 0.9 *^≠^	3.3 ± 0.4	3.6 ± 0.4	3.7 ± 0.8			
Δ	−0.1 ± 0.0	−0.1 ± 0.0	−0.1 ± 0.2	−0.1 ± 0.3	−0.1 ± 0.4			
Lo Lim MM-Pre (kg)	11.5 ± 1.7	11.9 ± 2.6	10.9 ± 1.7	12.1 ± 2.1	12.9 ± 1.9	0.300	0.414	0.108
Lo Lim MM-Post (kg)	11.4 ± 1.8	12.5 ± 2.8	11.4 ± 1.5	12.4 ± 2.3	12.6 ± 2.2			
Δ	−0.1 ± 0.1	0.6 ± 0.2	0.5 ± 0.2	0.3 ± 0.2	−0.3 ± 0.3			
ASM-Pre (kg)	15.1 ± 2.3	16.7 ± 3.4	14.6 ± 2.1	15.9 ± 2.6	16.3 ± 2.8	0.027	0.253	0.074
ASM-Post (kg)	14.8 ± 2.4	16.1 ± 3.3 ^#^	14.7 ± 1.9	15.8 ± 2.7	16.3 ± 2.9			
Δ	−0.3 ± 0.1	−0.6 ± 0.1	0.1 ± 0.1	−0.1 ± 0.1	0.0 ± 0.1			
ASMI-Pre (kg/m^2^)	6.2 ± 0.6	6.8 ± 0.9	6.1 ± 0.6	6.6 ± 0.7	6.7 ± 1.0	0.029	0.099	0.105
ASMI-Post (kg/m^2^)	6.1 ± 0.7	6.5 ± 0.9 ^#^	6.1 ± 0.5	6.5 ± 0.7	6.7 ± 1.0			
Δ	−0.1 ± 0.1	−0.3 ± 0.0	0.0 ± 0.1	−0.1 ± 0.0	0.0 ± 0.0			

CG: Control Group; ETG: Exergames Training Group; PSG: Protein Supplementation Group; ETPSG: Exergames Training combined with Protein Supplementation Group; ETISG: Exergames Training combined with Isoenergetic Supplementation Group; WC: waist circumference; Tot bod fat: total body fat; MM: muscle mass; Up lim: upper limbs; Lo lim: lower limbs; ASM: appendicular skeletal muscle mass; ASMI: appendicular skeletal muscle mass index; p^a^: *p*-values of within-group comparisons; p^b^: *p*-values between groups; p^c^: *p*-values of interaction; Δ: delta (difference between pre-and post-intervention assessments); * Significant difference interaction; ^#^:Significant difference within groups; ^≠^: Significant difference between groups; *n* = number.

**Table 4 ijerph-18-09324-t004:** Within- and between-group comparisons and interactions of gastrocnemius muscle (US) architecture.

Parametric Variables	CG (*n* = 15)	ETG (*n* = 14)	PSG (*n* = 16)	ETPSG (*n* = 14)	ETISG (*n* = 13)	p^a^	p^b^	p^c^
	Mean ± SD	Mean ± SD	Mean ± SD	Mean ± SD	Mean ± SD			
FL-Pre (cm)	3.0 ± 0.4	2.9 ± 0.4	2.9 ± 0.5	2.9 ± 0.3	3.3 ± 0.8			
FL-Post (cm)	3.0 ± 0.6	2.9 ± 0.4	2.8 ± 0.8	2.7 ± 0.5	3.2 ± 0.8	0.675	0.322	0.961
Δ	0.2 ± 0.2	0.0 ± 0.0	−0.1 ± 0.3	0.2 ± 0.2	−0.1 ± 0.0			
**Nonparametric Variables**	**CG (*n* = 15)**	**ETG (*n* = 14)**	**PSG (*n* = 16)**	**ETPSG (*n* = 14)**	**ETISG (*n* = 13)**	**p^a^**	**p^d^**	**p^e^**
	**Med (Min-Max)**	**Med (Min-Max)**	**Med (Min-Max)**	**Med (Min-Max)**	**Med (Min-Max)**			
MT-Pre (cm)	1.3 (1.1–1.8)	1.4 (1.1–1.7)	1.2 (1.0–1.6)	1.3 (1.1–1.7)	1.5 (1.1–1.8)			
MT-Post (cm)	1.4 (1.1–1.8)	1.4 (1.2–1.7)	1.4 (1.0–1.6)	1.3 (0.8–1.7)	1.4 (1.1–1.9)			
Δ	0.1 ± 0.2	0.0 ± 0.1	0.2 ± 0.1	0.0 ± 0.1	−0.1 ± 0.1	0.421	0.053	0.335
PA-Pre (°)	27.0 (23.0–35.0)	28.0 (24.0–36.0)	25.0 (18.0–32.0)	27.0 (21.0–30.0)	26.0 (24.0–33.0)			
PA-Post (°)	27.0 (20.0–36.0)	27.0 (22.0–41.0)	26.0 (21.0–32.0)	28.0 (25.0–32.0)	28.0 (20.0–31.0)			
Δ	0.0 ± 8.8	−1.0 ± 5.7	1.1 ± 4.4	1.0 ± 3.0	2.0 ± 3.5	0.878	0.081	0.444

CG: Control Group; ETG: Exergames Training Group; PSG: Protein Supplementation Group; ETPSG: Exergames Training combined with Protein Supplementation Group; ETISG: Exergames Training combined with Isoenergetic Supplementation Group; FL: fascicle length; MT: muscle thickness; PA: pennation angle; p^a^: *p*-values of within-group comparisons; p^b^: *p*-values between groups; p^c^: *p*-values of interaction; p^d^: *p*-values between-group comparisons (pre-experimental); p^e^: *p*-values of between-group comparisons (post-experimental); Δ: delta (difference between pre-and post-intervention assessments); Med: Median; Min: Minimum; Max: Maximum; *n* = number.

**Table 5 ijerph-18-09324-t005:** Within- and between-group comparisons and interactions of Plasma IL- (6) Levels, Ankle Peak Torque (PT), and Handgrip Strength (HS).

Variables	CG (*n* = 09)	ETG (*n* = 09)	PSG (*n* = 11)	ETPSG (*n* = 08)	ETISG (*n* = 05)	p^a^	p^b^	p^c^
IL-6-Pre (pg/mL)	2.8 ± 2.3	2.5 ± 1.1	3.5 ± 2.3	2.2 ± 0.7	4.2 ± 3.6	0.521	0.995	0.069
IL-6-Post (pg/mL)	3.1 ± 1.2	4.0 ± 3.7	2.7 ± 1.6	4.3 ± 2.8	2.6 ± 1.2			
Δ	0.3 ± 1.1	1.5 ± 2.6	−0.8 ± 0.7	2.1 ± 2.1	−1.6 ± 2.4			
	**CG (*n* = 15)**	**ETG (*n* = 15)**	**PSG (*n* = 17)**	**ETPSG (*n* = 16)**	**ETISG (*n* = 14)**			
HS-Pre (kg)	20.4 ± 5.7	20.1 ± 7.2	20.6 ± 4.1	20.3 ± 4.6	18.9 ± 7.1	0.008	0.858	0.222
HS-Post (kg)	20.1 ± 5.4	23.3 ± 6.2 ^#^	21.4 ± 3.6	21.4 ± 4.5	20.8 ± 3.6			
Δ	−0.3 ± 0.3	3.2 ± 1.0	0.8 ± 0.5	1.1 ± 0.1	1.9 ± 3.5			
PTplant 60°/s- Pre (Nm)	44.1 ± 8.8	38.2 ± 12.8	41.8 ± 16.9	37.1 ± 13.2	45.9 ± 18.5	1.566	0.395	0.735
PTplant 60°/s- Post (Nm)	44.3 ± 6.9	39.6 ± 12.1	40.7 ± 14.8	40.2 ± 11.9	48.0 ± 16.6			
Δ	−0.2 ± 1.9	1.4 ± 0.7	−1.1 ± 2.1	3.1 ± 1.3	2.1 ± 1.9			
PTdors 60°/s- Pre (Nm)	18.9 ± 2.9	18.7 ± 4.8	17.7 ± 2.9	16.3 ± 2.5	18.5 ± 4.6	0.031	0.359	0.472
PTdors 60°/s- Post (Nm)	19.7 ± 3.5	19.4 ± 4.3	17.6 ± 3.2	18.4 ± 4.2 ^#^	19.7 ± 5.3			
Δ	0.8 ± 0.6	0.7 ± 0.5	−0.1 ± 0.3	2.1 ± 1.7	1.2 ± 0.7			
PTplant 180°/s- Pre (Nm)	25.2 ± 6.4	22.9 ± 7.6	24.4 ± 9.8	20.8 ± 6.9	27.3 ± 10.0	0.113	0.085	0.256
PTplant180°/s- Post (Nm)	25.4 ± 5.6	19.5 ± 6.5	22.5 ± 7.7	20.4 ± 5.9	27.8 ± 10.1			
Δ	0.2 ± 0.8	−3.4 ± 1.1	−1.9 ± 2.1	−0.4 ± 1.0	0.5 ± 0.1			
PTdors 180°/s- Pre (Nm)	16.0 ± 3.2	17.1 ± 5.9	14.6 ± 2.2	14.7 ± 2.4	17.1 ± 5.3	0.853	0.342	0.687
PTdors 180°/s- Post (Nm)	16.3 ± 2.4	16.1 ± 3.2	14.8 ± 2.5	15.7 ± 3.9	16.2 ± 5.2			
Δ	0.3 ± 0.8	−1.0 ± 2.7	0.2 ± 0.3	1.0 ± 1.5	−0.9 ± 0.1			

CG: Control Group; ETG: Exergames Training Group; PSG: Protein Supplementation Group; ETPSG: Exergames Training combined with Protein Supplementation Group; ETISG: Exergames Training combined with Isoenergetic Supplementation Group; IL-(6): Interleukin 6; HS: Handgrip Strength; PT, Peak torque; Nm, Newton meter; °/s, degree/second; PT plant: plantar flexors peak torque; PT: peak torque of dorsiflexors; p^a^: *p*-values of within-group comparisons; p^b^: *p*-values between groups; p^c^: *p*-values of interaction; Δ: delta (difference between pre-and post-intervention assessments); ^#^: Significant difference within groups; *n* = number.

**Table 6 ijerph-18-09324-t006:** Comparison of frailty criteria within and between-group (*n* = 79).

Physical Frailty Criteria	CG (*n* = 15)			ETG (*n* = 15)		PSG (*n* = 18)		ETPSG (*n* = 16)		ETISG (*n* = 15)	p^a^	p^b^	p^c^
	Pré *n* (%)		p^a^	Pré *n* (%)	p^a^	Pré *n* (%)	p^a^	Pré *n* (%)	p^a^	Pré *n* (%)			
Fatigue/Exhaustion—Pre	5 (35.70)		0.688	7 (46.70)	0.016 ^≠^	8 (44.40)	0.031 ^≠^	7 (43.80)	0.016 ^≠^	8 (53.30)	0.125	0.823	0.187
Fatigue/Exhaustion—Post	3 (21.40)			0 (0.00)		2 (11.10)		0 (0.00)		3 (20.00)			
Weight loss—Pre	2 (14.30)		0.500	2 (13.30)	1.000	4 (22.20)	0.625	2 (12.50)	1.000	4 (26.70)	0.25	0.879	0.957
Weight loss—Post	0 (0.00)			1 (6.70)		2 (11.10)		1 (6.30)		1 (6.70)			
Low activity—Pre	1 (7.10)		1.000	0 (0.00)	NR	2 (11.10)	0.500	0 (0.00)	NR	0 (0.00)	-	0.504	0.179
Low activity—Post	1 (7.10)			0 (0.00)		0 (0.00)		0 (0.00)		0 (0.00)			
Low HS—Pre	9 (64.30)		0.625	8 (53.30)	0.125	11 (61.10)	0.219	9 (56.30)	0.625	8 (53.30)	1.000	0.879	0.376
Low HS—Post	7 (50.00)			3 (20.00)		7 (38.90)		7 (43.80)		8 (53.30)			
Low GS—Pre	0 (0.00)		NR	1 (6.70)	1.000	1 (5.60)	1.000	1 (6.30)	1.000	0 (0.00)	-	1.000	1.000
Low GS—Post	0 (0.00)			0 (0.00)		1 (5.60)		0 (0.00)		0 (0.00)			
Number of Physical Frailty													
Total Sum—Pre	1	10 (66.70)		11 (73.30)		11 (61.10)		12 (75.00)		11 (73.30)			
	2	05 (33.30)		04 (26.70)		07 (38.90)		04 (25.00)		04 (26.70)			
Total Sum—Post	0	07 (46.70)		11 (73.30)		10 (55.60)		07 (43.80)		05 (33.30)			
	1	05 (33.30)	0.016 ^≠^	04 (26.70)	0.001 ^≠^	05 (27.80)	0.000 ^≠^	09 (56.30)	0.063 ^≠^	09 (60.00)	0.016 ^≠^	0.862	0.347
	2	02 (13.30)		0 (0.00)		02 (11.10)		0 (0.00)		01 (6.70)			
	3	01 (6.70)		0 (0.00)		01 (5.60)		0 (0.00)		0 (0.00)			

NR, No Result; *n*, number; %, percentage; CG: Control Group; ETG: Exergame Training Group; PSG: Protein Supplementation group; ETPSG: Exergame Training and Protein Supplementation Group; ETISG: Exergame Training and Isoenergetic Supplementation Group; HS: Handgrip strength; GS: Gait speed; p^a^: within group; p^b^: between groups (pre-experiment); p^c^: between groups (post-experiment.); ^≠^: Significant difference within groups; *n* = number.

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
