# Peer review of "Effects of Exergames and Protein Supplementation on Body Composition and Musculoskeletal Function of Prefrail Community-Dwelling Older Women: A Randomized, Controlled Clinical Trial"

_ijerph, 2021, doi:10.3390/ijerph18179324_

Round 1

Reviewer 1 Report

It is necessary to explain how the intensity of the work performed (resistance and strength) has been controlled and adjusted.
Indicate which professional has designed and controlled the workload.

Reviewer 2 Report

- The authors investigated the effects of exergames and protein supplementation on body composition and musculoskeletal function of pre-frail older women. The authors claimed that the effect of this kind of multi-domain intervention on reducing frailty criteria or reversing pre-frailty status and improving musculoskeletal function in community-dwelling older people have not yet been investigated (L66-70), while they conducted similar study previously (ref #27). Please clearly describe what is the novelty of this study.

- The authors examined IL-6, while the aim of this assessment is not clear and there is no discussion on it throughout the MS.

- Please revisit L187 [31] and L190...

- Tables, their style is not good, hard to see.

- Abbreviations are complicated. Actually the authors didn't handle them. There is a lot of typo.

- There is a few discussion on the present findings. The authors mostly described the previous studies, while they didn't emphasize the significance of this study.

- The benefit of multi-domain intervention (i.e., ETPSG) is not clear as compared with CG.

- Please revisit L405-423. They are out of context.

- The adherence to the supplements may not so high. Please compare the adherence to other studies.

Reviewer 3 Report

Dear authors,

You present an interesting paper that contributes to the knowledge about the benefits of physical activity in physically frail older.

The study is interesting and provides some important information on how a multi-component interventions would improve body composition and musculoskeletal function.

Although the study has the potentiality of being shared with the scientific community, I believe that the manuscript would benefit from a minor revision with the attempt to better support your experimental setting.

  1. More information should be provided about the characteristics of the physical training program presented. The detailed training plan is important for the replication of this study.

  1. As also the authors stated one limitation of the study is that the fatigue/exhaustion criterion was assessed with two questions taken from Fried's Frailty Phenotype. In my opinion, more information should be provide about this method that seems to be insufficient.

  1. What were inclusion and exclusion criteria? Please, better define them.

  1. Why the physical training program was direct by physiotherapists? Wouldn’t it be more convenient ad appropriate to require the expertise of Motor sciences graduates?

  1. What was done to improve the adherence to the participants' intervention?

  1. I would like to see more of the practical implications, not only regarding the effects of this type of interventions but also in relation to the strategies for their implementation and characteristics that should be considered in the programs, based on the analyzed interventions.

Round 2

Reviewer 1 Report

The authors have improved their manuscript.

Reviewer 2 Report

- The authors conducted similar study previously (ref #27), and thus they should clearly describe what is the novelty of this study and what is the rationale to do this study in the Introduction and Discussion.

- This reviewer doesn't see the significance of this study. The data might not support the conclusion.

- The benefit of multi-domain intervention against frailty is very limited.
